# Inflammasomes in Alzheimer’s Progression: Nrf2 as a Preventive Target

**DOI:** 10.3390/antiox14020121

**Published:** 2025-01-21

**Authors:** Rubén López-Hernández, María Magdalena de la Torre-Álamo, Belén García-Bueno, Alberto Baroja-Mazo, Francisco Jose Fenoy, Santiago Cuevas

**Affiliations:** 1Molecular Inflammation Group, Pathophysiology of the Inflammation and Oxidative Stress Lab, Biomedical Research Institute of Murcia (IMIB), University Clinical Hospital Virgen de la Arrixaca, 30120 Murcia, Spain; rubeneslh@gmail.com; 2Molecular Inflammation Group, Digestive and Endocrine Surgery and Transplantation of Abdominal Organs, Biomedical Research Institute of Murcia (IMIB), University Clinical Hospital Virgen de la Arrixaca, 30120 Murcia, Spain; mariamagdalena.dlta@um.es (M.M.d.l.T.-Á.); belen.garcia@imib.es (B.G.-B.); alberto.baroja@ffis.es (A.B.-M.); 3Department of Physiology, Faculty of Medicine, University of Murcia, 30120 Murcia, Spain; fjfnoy@um.es

**Keywords:** Alzheimer’s, inflammasome NLRP3, Aβ proteins, ASC, inflammation, Nrf2

## Abstract

Current knowledge about Alzheimer’s disease highlights the accumulation of β-amyloid plaques (Aβ1–42) and neurofibrillary tangles composed of hyperphosphorylated Tau, which lead to the loss of neuronal connections. Microglial activation and the release of inflammatory mediators play a significant role in the progression of Alzheimer’s pathology. Recent advances have identified the involvement of inflammasomes, particularly NOD-like receptor NLR family pyrin domain containing 3 (NLRP3), whose activation promotes the release of proinflammatory cytokines and triggers pyroptosis, exacerbating neuroinflammation. Aggregates of Aβ1–42 and hyperphosphorylated Tau have been shown to activate these inflammasomes, while the apoptosis-associated speck-like protein (ASC) components form aggregates that further accelerate Aβ aggregation. Defects in the autophagic clearance of inflammasomes have also been implicated in Alzheimer’s disease, contributing to sustained inflammation. This review explores strategies to counteract inflammation in Alzheimer’s, emphasizing the degradation of ASC specks and the inhibition of NLRP3 inflammasome activation. Notably, the nuclear factor erythroid 2-related factor 2 (Nrf2) transcription factor emerges as a promising therapeutic target due to its dual role in mitigating oxidative stress and directly inhibiting NLRP3 inflammasome formation. By reducing inflammasome-driven inflammation, Nrf2 offers significant potential for addressing the neuroinflammatory aspects of Alzheimer’s disease.

## 1. Introduction

### 1.1. Alzheimer’s Disease

Alzheimer’s disease (AD) is a progressive neurodegenerative disease that primarily affects cognitive functions such as memory, thinking, and behavior. Some of the known risk factors include advanced age, family history of the disease, traumatic brain injury, hypertension, diabetes, and obesity. Two types of cases are differentiated in AD: those that occur sporadically (95%), and those that have a dominant inheritance pattern (5%). Although it is considered a senile disease, i.e., it normally affects older people, it can affect younger individuals due to various mutations, although the latter only account for 5% of Alzheimer’s cases [1].

According to figures provided on 15 March 2023 by the World Health Organization, currently more than 55 million people suffer from some type of dementia worldwide, being the seventh leading cause of death and one of the main causes of disability and dependence in the elderly population globally, with AD being the most common type of dementia [2]. Additionally, other cardiovascular pathologies such as hypertension, diabetes, and obesity are significant risk factors for AD due to their impact on cerebrovascular health, metabolism, and neuroinflammation. When combined, these conditions amplify each other’s negative effects on the brain. For example, diabetes and obesity worsen hypertension-related vascular damage, creating a vicious cycle of inflammation, oxidative stress, and neuronal injury that accelerates AD progression.

The discovery of this disease occurred in 1906 when the German physician and scientist, Alois Alzheimer, presented to the scientific community the disease that would later be known by his name. During the congress, Mr. Alzheimer presented the clinical case of Mrs. Auguste Deter, a 50-year-old patient who had been admitted to a hospital because of behavioral alterations reported by her spouse. After Auguste Deter’s death, Alzheimer requested brain tissue samples to perform an autopsy. Upon analysis, Alzheimer discovered that unusual protein deposits were found in and around the nerve cells. In addition, by means of silver staining he observed tangled bundles of fibrils at the sites of the neurons [3,4].

It is now known that there are two reasons for the onset of Alzheimer’s disease: the formation of senile plaques due to the accumulation of the protein β-amyloid (Aβ), and the formation of intracellular neurofibrillary tangles (NFTs) due to the accumulation of hyperphosphorylated Tau proteins. Amyloid plaques and neurofibrillary tangles mainly affect the hippocampus and prefrontal cortex, decreasing the density of cholinergic neurons, thus causing a cognitive and memory deficit.

The Aβ proteins play a central role in the disease due their accumulation in insoluble beta-sheet-rich fibrils. These fibrils disrupt neuronal communication by interfering with N-methyl-D-aspartate (NMDA) signaling, inhibiting glutamate transporters (EAAT) in supporting cells and overactivating ionotropic NMDA glutamate receptors, ultimately leading to excitotoxicity [5]. They also trigger sustained inflammatory responses, produce excessive free radicals that promote the generation of oxidative stress in nerve cells, and alter the flow of ions and other essential molecules within and around the cells [5].

This gradually develops a deficit of alpha-7-nicotinic acetylcholine receptor (α7nAChR), alterations in the transport of choline to the cortex and hippocampus, and alterations in the activities of acetyl transferase and cholinesterase enzymes of cholinergic neurons [6].

### 1.2. Aβ Proteins

Aβ proteins are peptides of about 36–43 amino acids with very diverse physiological functions; they participate against oxidative damage induced by transition metals [7], prevent cholesterol binding to low-density lipoproteins (LDL) when cholesterol levels are very high [8], and can modulate proinflammatory mechanisms [9], among other functions.

The Aβ protein is generated through the enzymatic cleavage of the precursor protein Aβ (APP). The amyloid precursor protein is a transmembrane protein that has multiple important functions in the cerebral cortex, including regulating synapses, neuronal plasticity, and iron export. This can come from three different mRNA species (*APP751*, *APP770*, and *APP695*) from an alternative splicing of the *APP* gene [10]. The *APP695* isoform is expressed only in neurons, whereas the *APP751* and *APP770* isoforms are expressed in both neurons and glia [9]. APP originates after translation of the *APP* gene, a highly conserved gene [11] comprising 18 exons [12] located in sub-band 3 of band 1 of region 2 of the long arm of chromosome 21 (21q21.3).

#### Molecular Mechanisms of Aβ Formation

The Aβ production process can follow two distinct pathways: the amyloidogenic pathway, and the non-amyloidogenic pathway.

In the non-amyloidogenic pathway, the α-secretase cleaves APP, releasing a soluble domain, sAPPα, into the extracellular medium, and leaving behind carboxy-terminal fragment CTFα or CTF83, bound to the plasma membrane. In this same non-amyloidogenic pathway, β-secretase, a transmembrane aspartyl protease, acts on the soluble sAAPα moiety, producing the soluble Aβ1-16 isoforms, while γ-secretase cleaves the carboxy-terminal fragment, CTF83, generating the membrane-bound APP intracellular domain (AICD) and soluble Aβ17-42/40 [13]. The activity of γ-secretase requires the binding of the catalytic subunit presenilin, which is encoded by either the presenilin 1 (*PSEN1*) or presenilin 2 (*PSEN2*) gene [14].

With respect to the amyloidogenic pathway, the first secretase to act is not α-secretase, but β-secretase; this implies, that on the one hand, a soluble sAPPβ fraction is formed with a membrane-bound fraction CTF99 subsequently with γ-secretase AICD and Aβ1–42/40. These Aβ1–42/40 isoforms would bind to metal ions, forming Aβ oligomers that would subsequently give rise to Aβ fibrils [13]. It has been observed in studies with transgenic mice that mutations in the *PSEN1* and *PSEN2* genes or in APP increase the proportion of Aβ1–42 over Aβ1-40 [13,15]. Mutations in the *PSEN1* gene usually result in more severe forms of the disease, while mutations in *PSEN2* are less aggressive, presenting incomplete penetrance.

Other evidence in relation to Aβ proteins that explain an increased likelihood of aggravating AD include variants in sortilin-related receptor 1 (*SORL1*), a gene that codes for an apolipoprotein E (APOE) receptor. This variant causes the receptor to be underexpressed and decreases Aβ clearance [16]. Variations in the APOE allele, specifically in ε4, where APOE4 is less effective at clearing amyloid plaques [17]. Variations in the IDE gene contribute to decreased degradation of Aβ protein [18]. Calcium homeostasis modulator 1 (*CALHM1*), a gene encoding a transmembrane glycoprotein that is involved in regulating the cytosolic Ca^2+^ concentrations and Aβ levels, has a P86L variant that interferes with Ca^2+^ permeability by increasing Aβ levels [19]. The addition of a pyroglutamate group at the third amino acid position on Aβ increases its ability to aggregate forming plaques [20].

### 1.3. Tau Proteins

Tau proteins are hydrophilic proteins with unfolded regions of about 352–441 amino acids in length with, 79 serine and threonine sites susceptible to phosphorylation; their function is to bind to the microtubules of axons, correctly maintaining neuronal projections and aiding axonal transport. They are encoded by the microtubule-associated protein Tau (MAPT) gene, located on the long arm of chromosome 17; the primary transcript has 16 exons, and alternative splicing mainly affects the N-terminal region and the microtubule-binding domain (MBD) [1,21].

MBD is extremely important in the binding of Tau to microtubules. In AD, it has been observed that hyperphosphorylations in Tau by proline-directed kinases such as glycogen synthase kinase-3 beta (GSK-3β) and cyclin dependent kinase 5 (CDK5) kinases induce Tau self-assembly in NFT [1], which prevents Tau binding to microtubules, leading to synaptic dysfunction [22] and its deposition in the form of NFTs, resulting in the formation of aggregates in the brainstem nuclei [22]. Specifically, GSK-3β is capable of phosphorylating Tau in the intracellular paired helical filaments of NFTs [1]. In the brains of patients with AD, GSK-3β colocalized with phosphorylated Tau [23] were found. It has also been observed that the activity of GSK-3β increases when the Aβ protein is present [1]. Ling Xie and colleagues demonstrated that Aβ acted as a direct competitive inhibitor of insulin-to-receptor binding [24], suggesting that Aβ prevents GSK-3β inhibition.

On the other hand, CDK5 is another kinase with Tau action that is activated after proteolysis from p35 to p25. The action of CDK5 kinase on Tau is capable of stimulating subsequent phosphorilazion by GSK-3β, probably by a dysregulation of CDK5 by p25 [1] (Figure 1).

When neuronal cells die, the NFTs aggregate further and are deposited in the extracellular milieu. While hyperphosphorylation and Tau deposition alone have not been shown to directly damage the cells, they hinder axonal transport. Over time, the lysosomes of neighboring cells attempt to degrade the NFTs, leading to chronic inflammation that ultimately triggers neuronal apoptosis [25].

### 1.4. Cooperation Between Aggregates of Aβ and Hyperphosphorylated Tau Give Rise to Alzheimer’s Disease

According to the amyloid hypothesis, Aβ would form a pore in the plasma membrane of neurons, allowing for the entry of extracellular calcium into the neurons and overactivating kinases involved in hyperphosphorylating Tau [20]. Aβ would internalize NMDA receptors through binding to α7 nicotinic receptors, reducing the frequency at the synapse [20].

Aβ can bind to extrasynaptic NMDA receptors, activating them and inducing calcium entry into the neuron, which in turn activates adenosine monophosphate-activated protein kinase (AMPK). Activated AMPK phosphorylates Tau in neuronal dendrites. This process not only triggers the separation of Tau from microtubules and its aggregation into NFT but also promotes an increased affinity of Tau for the Fyn protein, a non-receptor tyrosine kinase of the Src family kinases. Consequently, both Tau and Fyn migrate into the dendritic spine, significantly impacting proper neuronal function. [20]. Within the dendritic spine, Fyn kinase phosphorylates NMDA receptors, facilitating their coupling with the postsynaptic density protein 95 (PSD-95) protein. This connection is essential for Aβ peptide-associated toxicity to occur [20]. Binding Aβ to the cellular prion protein (PrPc) can also activate Fyn to phosphorylate NMDARs [19]. In more advanced stages of the disease, Aβ protein triggers the action of the phosphatase STEP enzyme on Fyn proteins. This leads to Fyn deactivation, which results in the loss of connections between neurons and the impairment of dendritic structures that receive neuronal signals [19]. This hypothesis suggests a cooperative role between Tau proteins and Aβ proteins in the initiation of AD [1,20,26].

### 1.5. Current Therapies and Rationale for the Work

AD has currently been the subject of intense pharmacological research aimed at finding effective treatments that can slow its progression or improve symptoms. Currently, the drugs approved for the treatment of AD focus on the management of cognitive and behavioral symptoms associated with cognitive and behavioral symptoms using reversible acetylcholinesterase (AChE) antagonists. Tacrine was the first drug to be approved for the dementia of AD [27,28], but is currently out of use due to its adverse hepatotoxic effects. Donepezil [29], rivastigmine [30], and galantamine [31], are commonly prescribed to raise acetylcholine (ACh) levels [32]. Other drugs employ different mechanisms of action, such as memantine, a weak non-competitive NMDA antagonist that decreases elevated glutamate levels by inhibiting NMDA receptor activation [33]. Another example is Aducanumab, which immobilizes Aβ clusters, facilitating their phagocytosis by the immune system [32,34] (Table 1).

Other therapeutic approaches currently under study for AD are as follows: Caprydilene, which improves cytoplasmic energetic capacity [35,36]; nonsteroidal anti-inflammatory drugs (NSAIDs) such as Flurbiprofen, which inhibits APP cleavage secretases [37]; and valproic acid, which blocks microtubule dissociation and has a neuroprotective effect in AD [38] rosiglitazone, which increases insulin sensitivity and glucose utilization [39,40], among others (Table 1). As mentioned earlier, AD is associated with hyperphosphorylated Aβ and Tau proteins in brain tissue. However, the progression of AD in patients is influenced by multiple factors, including pyroptosis mediated by inflammasomes. This leads us to question the extent of the relationship between these supramolecular complexes and AD, and whether existing treatments or potential therapies could modulate these complexes and significantly slow the progression of AD in affected patients.

**Table 1 antioxidants-14-00121-t001:** Current drugs used for the treatment of Alzheimer’s disease, and others being studied.

Drug	Mechanisms of Action	Current Use/State
Tacrine	AChE inhibitor.	In disuse due to hepatotoxicity [27,28].
Donepezil	AChE inhibitor.	Commonly prescribed to increase ACh levels [29].
Rivastigmine	AChE inhibitor.	Commonly prescribed to increase ACh levels [30].
Galantamine	AChE inhibitor.	Commonly prescribed to increase ACh levels [31].
Memantine	Weak non-competitive NMDA receptor antagonist; decreases elevated glutamate levels.	Used for moderate to severe Alzheimer’s disease [33].
Aducanumab	Monoclonal antibody that immobilizes Aβ clumps, allowing increased phagocytosis by the immune system.	Approved by the US Food and Drug Administration (FDA) for the treatment of early Alzheimer’s disease [34]. However, its use is controversial [40].
Caprylidene	Improves cytoplasmic energy capacity.	In reference [35].
Flurbiprofen	NSAID inhibiting APP-splitting secretases.	In reference [37].
Valproic acid	Blocks microtubule dissociation.	In reference [38].
Rosiglitazone	Increases insulin sensitivity and glucose utilization.	In reference [40].

## 2. Alzheimer’s Disease and the Immune System

The action of the immune system is known to play a crucial role in the development of AD, particularly in the inflammatory process. When Aβ deposition increases significantly, proinflammatory systems are activated, leading to chronic inflammation that affects the brain. This process results in increased oxidative stress, characterized by the production of reactive oxygen species (ROS) and nitric oxide (NO), as well as ischemia in neural pathways, which reduces oxygen utilization by neurons [41].

The first immune cells to detect Aβ deposition are microglia, phagocytic cells that are part of the central nervous system (CNS) and act as active sensors of the brain environment, constantly monitoring for pathogens or harmful agents. The detection of Aβ accumulations by microglia, along with astrocytes, triggers the production of cytokines such as fibroblast growth factor 2 (FGF2), interleukin 6 (IL-6), tumor necrosis factor-alpha (TNFα), and interleukin 1 (IL-1). These cytokines promote a proinflammatory state that increases the permeability of the blood–brain barrier (BBB), allowing for the entry of other phagocytic cells into the CNS. Macrophages and dendritic cells are thought to play a critical role in aiding the removal of Aβ [42]. However, since Aβ production is continuous, the neuroinflammatory state persists, causing the BBB to consistently allow for the entry of large molecular weight molecules and cells. This includes both common and uncommon bacteria and viruses, which further aggravate the overall condition of the brain. Additionally, it is known that the BBB in the hippocampus is particularly susceptible to deterioration with age [43]. This suggests that this may be one of the factors contributing to the aggravation of AD, particularly in the hippocampus. Accumulations of Aβ, along with cytokines, glutamate, noradrenaline, and other molecules, lead to reactive astrogliosis, which exacerbates neuroinflammation and neurotoxicity [44]. Receptors expressed on immune cells play a crucial role in regulating inflammation. In particular, the triggering receptor expressed on myeloid cells 2 (TREM2), which is expressed on microglia, has been shown to contribute to anti-inflammatory functions in the brain. Variants of this receptor are associated with an increased risk of developing certain conditions [45].

The immune system not only includes mechanisms capable of eliminating pathogens or harmful molecules such as Aβ, but also encompasses systems that can directly affect the viability of our own cells. When pathogens or cellular damage are present, immune cells are capable of activating programmed cell death (PCD). There are several types of PCD such as apoptosis, necroptosis, PANoptosis, and pyroptosis. Some of them, such as apoptosis, are not directly related to the inflammatory process, while others, like pyroptosis, are [46].

### 2.1. A Role of the Immune System in Alzheimer’s Disease, the Inflammasome

Pyroptosis is a natural antimicrobial mechanism that typically occurs in inflammatory environments during an invasion by intracellular pathogens. This type of cell death is highly effective at destroying the replication niche of the hereditary material of intracellular pathogens, while also attracting other immune cells to the site of damage. This process is facilitated by the inflammasome [47].

The inflammasome is a large supramolecular complex responsible for initiating the lytic cycle characteristic of pyroptosis. It is formed when specific damage- or pathogen-associated molecular patterns (DAMPs and PAMPs, respectively) interact with pattern recognition receptors (PRRs) on the cell’s plasma membrane. These PRRs include nucleotide-binding oligomerization domain-like receptors with leucine-rich repeats (NLRs), absent in melanoma 2 (AIM2) and IFN-inducible protein 16 (IFI16), among others. The inflammasome also includes PRRs that can activate it, although they are not always required. For inflammasome activation, certain PRRs, such as NLRs, can cooperate with pre-stimulatory signals from toll-like receptors (TLRs) [48]. Inflammasomes are defined by specific members of the NOD-like receptor (NLR) family.

During inflammasome assembly, an NLR oligomerizes with apoptosis-associated speck-like proteins (ASCs) through homotypic interactions between the pyrin domain (PYD) of the ASC and the pyrin domain of the NLR [49]. Assembly of the inflammasome is completed when pro-caspase-1 is recruited to the caspase recruitment domain (CARD) presented by the ASC adaptor protein. This binding enables the autocatalytic cleavage of the pro-caspase-1 zymogen into its active form, caspase-1. Active caspase-1 can then cleave cytosolic pro-interleukin-1β (pro-IL-1β) and pro-interleukin-18 (pro-IL-18) into their mature forms, IL-1β and IL-18, respectively, which are capable of promoting inflammation. [50] (Figure 2).

Specifically, IL-1β interacts with its receptor and transmits signals in a manner similar to toll-like receptors (TLRs), as it also contains a TIR domain. These signals activate nuclear factor kappa B (NF-κB) through the myeloid differentiation primary response protein 88 (MyD88), leading to the activation of proinflammatory genes such as IL-6 and TNF-α. On the other hand, IL-18 promotes vascular components of inflammation, including the expression of cell adhesion proteins and the production of IFN-γ [51].

Caspase also produces a proteolytic cleavage of gasdermin D (GSDMD) by removing the inhibition of the C-terminal domain (GSDMD-CT) of gasdermin D (GSDMD-CT) at the N-terminus (GSDMD-NT) [52]. This enables GSDMD to bind to the plasma membrane phospholipids and form pores with an inner diameter of 10–14 nm, through which IL-1β and IL-18, along with various cytosolic molecules critical for proper cell function, are released into the extracellular space. This release halts normal cellular activities and triggers a massive influx of water into the cell, leading to increased osmotic pressure. Ultimately, this culminates in the final phase of pyroptosis: litic cell death [53] (Figure 2).

There are two pathways of inflammasome activation: a canonical pathway, and a non-canonical pathway. The above corresponds to inflammasome activation by a canonical pathway. This is because it involves the three classical components: an NLR, ASC, and procaspase-1. In a non-canonical pathway, this pathway starts with pro-caspase 4 and 5 or pro-caspase 11 in humans and mice, respectively [54]. These caspases are able to detect intracellular bacterial lipopolysaccharide (LPS), oligomerize, cleave to their mature forms, and cleave GSDMD themselves, creating the lysis pore [55,56].

Pyroptosis generally plays a beneficial role in the body’s defense against pathogen invasion by limiting the replication and spread of microorganisms. However, when pyroptosis occurs in an excessive or uncontrolled manner, it can trigger an intense and immoderate inflammatory response, exacerbating tissue damage and contributing to the pathogenesis of various inflammatory and autoimmune diseases. In this regard, precise regulation of pyroptosis is crucial to maintain the balance between elimination of pathogens and prevention of excessive damage to host tissues [57].

One of the best-studied inflammasomes is NLR family pyrin domain containing 3 NLRP3 (also known as NALP3 or cryopyrin), which is expressed mainly in myeloid cells such as macrophages and dendritic cells. It is activated specially in response to DAMPs and is able to do so in the absence of pathogens. These molecular patterns, such as cytosolic components found in the extracellular space (where they should not be found), have the ability to activate NLRP3 and report tissue damage. NLRP3 activation by DAMPs is a crucial mechanism in the inflammatory response under sterile conditions. This process is essential for eliminating damaged cells and initiating repair mechanisms. However, if this response becomes deregulated, it can lead to chronic inflammation and tissue damage. Examples of DAMPs include extracellular adenosine triphosphate (ATP), monosodium urate, environmental irritants, cholesterol crystals, and Aβ proteins. This mechanism is especially important in inflammation occurring in non-infectious conditions, pointing to it as a key molecular target to reduce pathological inflammation associated with diseases such as AD [58] (Figure 2).

In AD, the activation of the inflammasome, particularly the NLRP3 inflammasome, plays a pivotal role in the exacerbation of neuroinflammation and neuronal pyroptosis. In recent years, pyroptosis has been associated with AD progression [59,60] by activating NLRP1, AIM2, and NLRP3 inflammasomes through their interaction with Aβ proteins and hyperphosphorylated Tau proteins [59]. The persistent activation of NLRP3 in response to Aβ accumulation leads to the release of proinflammatory cytokines like IL-1β and IL-18, which contribute to neuronal damage and death [61]. This activation is tightly linked to cognitive decline, as the inflammatory cascade impairs synaptic function and plasticity, critical for learning and memory. Moreover, the pyroptotic cell death of neurons, induced by inflammasome-mediated processes, exacerbates the loss of neuronal integrity, further impairing cognitive processes [62]. As neurons undergo pyroptosis, they release DAMPs, which perpetuate the inflammatory cycle, further amplifying neuroinflammation and accelerating the decline in cognitive function typical of AD [63]. This link between inflammasome activation, pyroptosis, and cognitive decline underscores the importance of targeting the NLRP3 inflammasome to develop potential therapeutic strategies for AD [64].

### 2.2. Pyroptosis Is Induced by Aβ Accumulation

A study conducted by Nanjing University, China, demonstrated that Aβ 1–42 deposits induced pyroptosis in mouse cortical neurons (MCNs). The study included a negative control group, a positive control group (in which MCNs were treated with LPS and Nigericin to induce pyroptosis), and an experimental group exposed to Aβ [65]. Nigericin is a chemical compound that facilitates the exchange of hydrogen ions (H+) across cell membranes, leading to an increase in intracellular pH and the breakdown of pH gradients across membranes [66]. This can have several effects on cells, such as altering ionic homeostasis, releasing neurotransmitters, and inducing programmed cell death, as may be the case in pyroptosis [65].

LPS is a component of the cell wall of Gram-negative bacteria that also has the potential to trigger pyroptosis in cells [67]. Cell permeability was found to be higher in the Aβ 1–42 group compared to the negative control group, coming much closer to the positive control group. Propidium iodide (PI) was used to measure cell permeability [65]. The mRNA levels of GSDMD, a key transcript for the formation of the characteristic pyroptosis pore, were also higher than in the negative control group [65]. An increase in proinflammatory proteins typical of pyroptosis was observed in the Aβ 1–42 group. An enzyme-linked immunosorbent assay (ELISA) was used for their detection [65]. The expression levels of GSDMD, caspase-1, and NLRP3, among others, were also up-regulated in the Aβ 1–42 group [65]. The study also looked at the effect of inhibitors of key molecules in pyroptosis such as caspase-1 or GSDMD [65].

In addition, the cognitive impairment of APP/PS1 double transgenic mice (AD model mice) in which caspase-1 expression was silenced was analyzed. Threated mice showed a significant decrease in IL-6, IL-1β, and TNF-α and less nerve cell damage than control APP/PS1 mice [65].

As mentioned above, NLRP3 is the most studied one and the one that seems to be most related to pyroptosis in Alzheimer’s disease, but other inflammasomes have also been reported to contribute to the disease, such as NLRP1. In a study conducted by M-S Tan and colleagues [68] they showed that Aβ 1–42 aggregates increase NLRP1 expression in mice that were 6 months old. They also measured levels of NeuN, a neuronal nuclear antigen that acts as a marker of the functional status of neurons. The results show that NeuN levels were significantly reduced. They used dual immunofluorescence staining to identify the colocalization of NLRP1 with the neuron-specific marker NeuN. They also found thatsilencing of NLRP1 by siRNA reduced IL-1β secretion and lactate dehydrogenase release and led to increased cell densities in the cortex and hippocampus of mice. As in the previous study, the team tested caspase-1 knockdown with caspase-1 siRNA in 5-month-old APP/PS1 mice using non-viral RNA interference methodology in vivo for 6 weeks. The treated APP/PS1 mice had an improvement in cognitive performance, a reduction in the density of positive cells with the TUNEL technique, and an increase in the density of Nissl-positive cells in the cortex and hippocampus [68]. It seems obvious that Aβ 1–42 is capable of inducing pyroptosis in neuronal cells, and this raises the question of the mechanisms by which this happens.

Toll-like receptors such as TLR2 and TLR4 have been shown to trigger an inflammatory response to Aβ aggregates [69] and TLR4 is able to activate the NLRP3 inflammasome in mouse microglia [70]. Application of CLI-095, a TLR4 inhibitor, shows how, in a Western blot analysis, it down-regulates NLRP3, ASC, and Caspase-1 p10 protein levels in mouse microglia [70]. The sequences of the Aβ protein itself can act as a DAMP on inflammasomes. First, a cell membrane-specific receptor, such as a TLR, detects Aβ protein sequences, leading to the recruitment of IL-1 receptor-associated protein kinases (IRAKs) by MyD88. Upon phosphorylation, IRAKs activate TNF receptor-associated factor 6 (TRAF6). TRAF6, in cooperation with TAK1 and TAB kinases, phosphorylates IKK and, subsequently, IκB. This results in the activation of NF-κB, which initiates the transcription of inflammasome components [71] (Figure 2).

Second, microglia-mediated phagocytosis leads to the interaction of Aβ DAMP sequences with the inflammasome receptor, which activates caspase-1 via autocatalysis, resulting in the formation of IL-1β and IL-18, cleavage of GSDMD, and the eventual formation of the lysis pore [71]. Additionally, a study by the University of Massachusetts demonstrated that phagocytosis of Aβ by microglia caused lysosomal damage, stimulating the release of Cathepsin B into the cytoplasm [72]. Cathepsin B is a cysteine protease with intracellular activity linked to pathological conditions [73], and has been previously associated with AD [72]. This release of Cathepsin B induced microglial release of IL-1β [74]. Furthermore, Cathepsin B inhibitors, such as CA074 methyl ester, were shown to effectively inhibit IL-1β production and pyroptosis [74].

### 2.3. ASC Specks and Their Link to Alzheimer’s Disease

Another key factor in inflammasome activation, recently discovered, is the so-called ASC speck. These are irregular, helical, fibrillar aggregates of ASCs within the inflammasome that form when ASCs oligomerize into large, insoluble aggregates [71]. ASC specks are released from cells into the extracellular space during pyroptosis and are subsequently internalized by neighboring cells. They have the ability to amplify signaling responses that enhance caspase-1-mediated cytokine maturation, thereby increasing inflammatory activity. This is because the rapid oligomerization of ASCs through their PYD creates numerous potential sites for caspase-1 activation [75]. These ASC specks are subjected to an autophagy pathway for degradation inside the cell. However, for reasons that are still unknown, this system can fail [76]. When pyroptosis occurs in a microglial cell, ASC specks can be released into the extracellular space, where neighboring microglial cells may internalize them. If these cells fail to degrade the specks, they activate pyroptosis and release these fibrillar aggregates again, thereby creating a positive feedback loop that exacerbates the inflammatory response in AD [71] (Figure 3).

One of the possible reasons why this happens could be the phagocytic inability of microglia on Aβ, thus attempting to recruit other phagocytic cells to the site of the clusters in order to have greater phagocytic capacity [77]. In addition, it has been observed that extracellular speck ASCs would exhibit prion-like activities, being able to aggregate to cytosolic soluble ASCs upon escape from phagosomes, thus promoting their polymerization [78]. These speck ASCs, like prions, are highly resistant to extracellular proteases, which means that they remain for a long time in the extracellular space, overactivating the immune response in inflammatory diseases [78].

It is also known that extracellular ASC specks are capable of activating Caspase-1 in the same way as cytosolic ASC specks do; however, the role of extracellular caspase-1 in AD is still not well understood [79]. The role of ASC specks in AD is strengthened by the fact that ASC specks promote Aβ aggregation [80]. In this study, ASC specks were observed to bind to the first 42 amino acids of extracellular Aβ. Purified ASC specks co-incubated with Aβ1–42 accelerated Aβ aggregation [80].

The interaction of ASC specks with Aβ was produced by the PYD of ASC, and it was found that different mutations in the PYD of ASC completely prevent the promoter effect of ASC speck on Aβ aggregation, whereas mutations in the CARD do not prevent ASC speck from acting on Aβ aggregation [80].

Thus, these ASC specks, due to their contribution to the inflammatory process occurring in AD, present a potential therapeutic target for the development of new treatment alternatives in AD. In a 2023 study [81], a team of scientists developed IC 100, a humanized monoclonal antibody (IgG4k) that interferes with ASC polymerization and ASC speck formation, successfully reducing neuroinflammatory effects in certain pathologies. In another study of 2024 [82] was demonstrated the utility of ASC specks as a fluid biomarker of EA. With blood being the main focus for further development as convenient sample for diagnostics and clinical trials. The results of these recent studies open the door to studying ASC specks as a therapeutic target in AD.

### 2.4. Post-Translational Modifications Regulate NLRP3 Activation and ASC Agglomeration

Post-translational modifications (PTMs) are fundamental processes that occur after a protein has been synthesized and are crucial for diversifying functionality and regulating its activity in cells. These modifications can include phosphorylation, glycosylation, and acetylation, among others, and can alter the structure, subcellular localization, and interaction of proteins with other molecules, as well as regulate their stability and half-life [83]. Inflammasomes are no exception, as they are also susceptible to a multitude of post-translational modifications that allow for them to modulate their inflammatory responses to infection or tissue injury. Specifically, inflammasomes have specific regions in which their phosphorylation is able to regulate their activation [84].

Previous studies analyzed which sites were susceptible to post-translational phosphorylation of NLRP3 using a murine cell line retrovirally expressing NLRP3-FLAG. After purifying NLRP3-FLAG, the complex was subjected to trypsin, chymotrypsin, and Glu-C to obtain various peptides. These peptides were then localized using reverse-phase liquid chromatography followed by mass spectrometry to identify the NLRP3 phosphorylation sites [85].

All 3 sites found were serine regions. By mutating the nucleotide sequences that give rise to these amino acids, they observed that mutations in S161 or S728 lead to activity similar to the wild type (WT) form of NLRP3. However, they found that changes in the S5 amino acid resulted in reduced NLRP3 activity by decreasing the formation of ASC specks. Specifically, ASC speck formation was significantly reduced when S5 was changed to alanine and disappeared completely when S5 was mutated to phosphomimetic aspartate [85]. The study suggests that mutating S5 with amino acids that allow for it to give a negative charge by phosphorylation disrupts the interaction between the PYD of NLRP3 and the PYD of ASC. In other words, NLRP3 activation can be regulated by the loading of a single amino acid of the PYD [85].

Other studies show how different enzymes are able to dephosphorylate sites critical for inflammasome activation, such as non-receptor tyrosine phosphatase 22 (PTPN22), which dephosphorylates NLRP3 at tyrosine Tyr861. In contrast, PTPN22 does not appear to act on NLR family CARD containing 4 (NLRC4) or AIM2 inflammasomes, suggesting that it specifically modulates NLRP3 [86].

Other enzymes such as NLRP3 protein kinase A are able to phosphorylate at Ser295, attenuating NLRP3 ATPase activity and causing its inhibition [87] and at Ser291, also preventing activation and leading NLRP3 to degradation via ubiquitination [88]. These findings emphasize the interplay between phosphorylation and protein degradation pathways in the regulation of inflammasome activation, highlighting how it can integrate multiple signaling cascades to ensure appropriate cellular responses to inflammatory stimuli.

In conclusion, PTMs play a crucial role in modulating inflammasome activity and its interactions with various cellular signaling pathways, such as apoptosis, autophagy, and NF-κB signaling. These modifications help maintain a balance between effective pathogen defense and tissue damage [89]. In neurodegenerative diseases like Alzheimer’s disease, dysregulation of PTMs contributes to chronic inflammation, neuronal death, and cognitive decline [90]. Therefore, understanding how PTMs influence inflammasome activation and their crosstalk with other cellular pathways is essential for developing therapeutic strategies aimed at controlling neuroinflammation and mitigating disease progression.

## 3. Potential Inhibitory Treatments for Inflammasome Activity in AD

As mentioned above, silencing the expression or inhibition of crucial molecules in inflammasome assembly and activation could be therapeutic targets to reduce the toxic effects of inflammation in AD. In this section, we will present studies demonstrating how different molecules inhibit the action of inflammasomes that act negatively in AD (Table 2).

### 3.1. Antibodies Directed Against ASC Specks

While the body’s own mechanisms for degradation of ASC specks may fail, research continues to investigate whether there are other mechanisms that can degrade these aggregates more efficiently and whether we can induce them artificially. We are well aware that the use of antibodies can elicit a wide range of responses from the immune system, so the search for specific antibodies against ASC specks could lead to the discovery of an effective treatment for these aggregates. In one part of the study [80], in which they discussed how ASC specks promote Aβ aggregation, they used anti-ASC speck antibodies in APP/PS1 mice, specifically specific IgG that prevented ASC speck-induced Aβ aggregation in a concentration-dependent manner. APP expression, together with its cleavage products and Aβ aggregation per se, were not affected by the antibodies.

As mentioned earlier in Section 2.3, in a recent study [81], a team of scientists developed IC 100, a humanized monoclonal antibody (IgG4κ) that interferes with ASC polymerization and ASC speck assembly. They observed that, at concentrations of 1.0 and 0.1 μg/mL of IC100, IL-β release dropped significantly by inhibiting inflammasome activation. IC100 is first internalized into various cell types; from here, it can either be taken up by endosomes and released into the extracellular milieu, or associate in a complex with ASC and tripartite motif-containing protein 21 (TRIM21). This complex inhibits the activity of the proteosome on ASC, allowing for it to remain free in the cytoplasm for at least 3 days. The uptake of ASC by IC100 prevents the inflammasome from polymerizing with ASC and forming ASC specks [81]. IC100 is known to reduce neuroinflammatory effects in some pathologies [99], but there is no evidence yet that IC100 improves cognitive effects in AD, although it is used for the identification of inflammasome signaling proteins in AD [94].

### 3.2. NLRP3 Inhibitors in AD

#### 3.2.1. The Ketone Body β-Hydroxybutyrate Inhibits NLRP3 Inflammasome-Mediated Inflammation

β-hydroxybutyrate (BHB) is a ketone body product of the breakdown of fats in the liver that are produced in the absence of sufficient glucose needed by the body. These compounds, which include acetone, acetoacetate, and BHB, are not only an alternative source of energy for the body during periods of fasting or low-carbohydrate diets, but also have beneficial effects on modulating inflammation, reducing oxidative stress [100], etc.

In a study by Yun-Hee Youm [91] and colleagues, they found that β-hydroxybutyrate had inhibitory capacity on NLRP3 in mouse bone marrow-derived macrophages. This was possible because BHB was able to inhibit ATP-induced cleavage of caspase-1 on p20 and prevent processing of the biologically active p17 form of IL-1β at BHB concentrations similar to those induced by intense exercise or 2 days of fasting [91], and BHB was found to impair the decrease in intracellular K^+^ flux in macrophages, an important factor for NLRP3 activation [92].

As mentioned above, amyloid plaques had the ability to induce the oligomerization of ASCs by increasing NLRP3 activation and pyroptosis. This same study shows how BHB is able to prevent ASC oligomerization and the appearance of ASC specks [91]. Another study shows how BHB is at very low blood levels in AD patients and how BHB has the potential to reduce amyloid plaques and microgliosis in 5XFAD mice by reducing microgliosis, the amount of ASC specks, and the activity of inflammasomes [93]. This finding suggests that administration of β-hydroxybutyrate without restricting glucose intake could be a promising area of research for the reduction in inflammation in AD.

#### 3.2.2. Protective Effect of MCC50 in AD

MCC950, an inhibitor of NLRP3 activation, has emerged as a promising compound in medical research due to its ability to selectively block NLRP3-mediated inflammatory signaling.

The use of MC950 at nanomolar concentrations is able to specifically inhibit NLRP3-dependent pyroptosis, affecting both canonical and non-canonical NLRP3 activation in mouse bone marrow macrophages [95]. Specifically, MCC950 inhibits procaspase-1 cleavage, decreasing pro-IL-1β processing [95]. ASC speck formation is also attenuated by MCC950 at cell-line-dependent concentrations [95]. The intracellular K efflux in macrophages was not inhibited by MCC950, suggesting that MCC950 inhibits NLRP3 activation downstream of K^+^ efflux [95]. Other studies [96] showed that the inhibitor MCC950 had beneficial effects in APP/PS1 mice. A significant reduction in Aβ plaque burden was observed, as well as a decrease in brain inflammation and an improvement in cognitive function after MCC950 treatment [96]. In APP/PS1 mice treated with MCC950, a decrease in IL-1β concentration was also observed [96].

Tau proteins, also key in the pathogenesis of AD, are able to activate NLRP3, as demonstrated in a study by the University of Hasselt, Belgium [97]. Loss of NLRP3 inflammasome function in Pycard (Tau22/Asc-/-)-deficient mice reduces Tau hyperphosphorylation by regulating Tau kinases and phosphatases, in particular CaMKII-α, which phosphorylates Tau at Ser416 [101]. In transgenic mice lacking ASC or with the inhibitor MCC950, a marked decrease in the development of exogenous seeding-induced Tau pathology is observed [97].

#### 3.2.3. JC-124, an Effective Inhibitor of the NLRP3 Inflammasome in AD

JC-124 is a new compound with the potential to alleviate cognitive deficits in AD [98]. It is able to block NLRP3 formation and caspase-1 activation, thereby reducing IL-1β production in both in vitro studies and in living organisms. JC-124 was tested in 9-month-old RND8 APP (TgCRND8) transgenic mice for one month, which have increased expression of NLRP3, ASC, and cleaved caspase-1 [98].

JC-124-treated mice were observed to have significantly smaller amyloid plaques in the cortex than untreated mice. The deposition of Aβ oligomers in the cortex and hippocampus was also significantly reduced. Levels of β-CTF or CTF99 were reduced in JC-124-treated mice, so this compound was able to inhibit β-secretase, a key protein in the cleavage of APP in the amyloidogenic pathway that promotes the formation of Aβ1–42/40. Low levels of heme-oxygenase 1 (HO-1), an antioxidant enzyme, and 4-Hydroxynonenal (HNE), a toxic aldehyde formed as a product of cell membrane lipid peroxidation, revealed that JC-124 was also able to reduce oxidative stress in TgCRND8 mice [98].

## 4. Nrf2: A Crucial Inhibitory Mechanism in AD

Nrf2 (nuclear factor erythroid 2-related factor 2) is a leucine zip transcription factor that is activated under conditions of oxidative stress in AD [102]. Under homeostatic conditions, Kelch-like ECH-associated protein 1 (Keap1) negatively regulates Nrf2 activity by ubiquitination [103,104], preventing it from translocating to the nucleus and binding to the antioxidant response element (ARE). When cells face oxidative stress, certain cysteine residues in the Keap1 protein (containing sulfhydryl or -SH groups) undergo chemical modifications. These modifications change the structure of Keap1, reducing its affinity for Nrf2 and allowing for Nrf2 to be released.

Once Nrf2 is free of Keap1, it can translocate to the cell nucleus, where it binds to ARE regions of DNA and activates transcription of antioxidant genes [103]. This triggers a response that helps neutralize the damaging effects of oxidative stress. Nrf2 not only activates its target genes via ARE, but also limits the transcriptional activation of certain genes, whether or not they contain ARE, for example, by blocking the recruitment of RNA Polymerase II leading to transcriptional impairment of the cytokines IL-6 and IL-1β [105]. Nrf2 also inhibits β-secretase expression and Aβ production [106]. Activation of Nrf2 by methysticin has been shown to reduce neuroinflammation, hippocampal oxidative damage, and memory loss in APP/Psen1 mice [107]. However, Nrf2 activation has been found to be low in cells of neuronal origin, and neurons show a high capacity to degrade Nrf2 due to the abundant expression of cullin 3 proteins [103,108]. Furthermore, at advanced ages, the ability of Nrf2 to be activated decreases, leading to reduced cellular resistance against oxidative damage [109,110]. Initially, the accumulation of Aβ1–42 causes an increase in Nrf2 levels, reflecting a protective cellular response to ROS, but up-regulation of Keap1 also by Aβ1–42 ultimately blocks Nrf2 activity, thus contributing to the progression of AD [103].

Although the mechanisms have not yet been elucidated, it is now known that ROS is able to activate the inflammasome in mice with deficient expression of superoxide dismutase 1 (a major producer of ROS) NLRP3 is inhibited [111]. The results of a study by Xiuting Liu and colleagues [112] established a link between the Nrf2-ARE pathway and the NLRP3 inflammasome; activation of Nrf2 induces the expression of NADPH deshydrogenase quinone 1 (NQO1), which inhibits the formation of the NLRP3 inflammasome. Compounds such as tert-butylhydroquinone, dimethyl fumarate, epigallocatechin-3-gallate, citral, biochanin, or mangiferin up-regulate Nrf2 and inhibit NLRP3 [113].

### Nrf2 Activators as Therapeutics in Neuronal Inflammation

Compounds capable of increasing Nrf2 expression have been reported in recent years and appear to show promise in reducing inflammation in neurodegenerative diseases (Table 3). An endogenous antioxidant, DJ-1, which has pathogenic mutations associated with Parkinson’s disease, acts as an oxidative stress sensor suppressing the progression of several neurodegenerative disorders. DJ-1 contains three cysteine residues that function as sensors of oxidative stress; the activity of DJ-1 specifically targets the oxidation of residue C106, specifically the sulfenated form (-SOH) and its sulfinate form (-SO2H), appearing to have the highest antioxidant efficacies. The efficacy of DJ-1 was tested with the addition of compound B (able to bind to the -SO2H form and maintain its reduced form) and showed an improvement in spatial memory in APP/PS1 mice when repeatedly administered intraperitoneally [114]. DJ-1 positively regulates Nrf2 transcription [115], and compound B, which binds to DJ-1, potentiates Nrf2 activation through the PI3K/Akt pathway [116]. ND-13, a peptide derived from the most conserved sequence of DJ-1 that also led to the activation of the Nrf2 pathway, was demonstrated to be effective in protecting neuronal cultures from the effects of relevant neurotoxins in Parkinson’s disease and other neurodegenerative diseases [117]. In addition, ND-13 exerts protective effects by reducing apoptosis and reactive oxygen species accumulation and by inactivating the proapoptotic protein caspase-3 in neuronal cell lines exposed to these neurotoxic insults [118].

Another compound, omaveloxolone (RTA408), has obtain recently the FDA approval for the treatment of Friedreich’s ataxia [119]. RTA408 has recently obtained promising results, restoring damaged hippocampal neurons and decreasing Aβ1–42 aggregates in APP/PS1 mice treated with glutamate (inducer of oxidative stress and mitochondrial damage). RTA408 specifically was able to increase the expression of Nrf2 and NQO1 [120]. Another study suggests that Dl-3-N-Butylphthalide is able to inhibit NLRP3 by suppressing thioredoxin-interacting protein interactions with NLRP3 [121]. Recently, with the approval of dimethyl fumarate (a compound capable of blocking the interaction between Keap1 and Nrf2) for the treatment of multiple sclerosis, other studies [122] have suggested that the use of certain skin allergens, such as dimethyl fumarate, phthalic anhydride, or methyl heptine carbonate, at safe concentrations could yield promising results in AD by decreasing ROS, inhibiting inflammatory activity, or even by aiding intracellular Ca^2+^ homeostasis, as it has also been suggested that up-regulation of Ca^2+^ may be upstream of the mechanisms that produce AD [123].

Sulforaphane, an isothiocyanate produced when cruciferous vegetables are chewed, is another compound that prevents oxidative stress-induced cytotoxicity through the increase in the Nrf2 expression and decreasing HO-1 and NQO-1 targets in neuro2a cells and the sciatic nerve of diabetic animals, thus demonstrating its protective potential in neurodegenerative diseases [124]. The antioxidant and anti-inflammatory effects in Resveratrol, a chemical found mainly in red grapes, inhibits the aggregation of Aβ proteins and modulates intracellular effectors involved in neuronal survival/death [125].

Continuing other plant-derived compounds, ellagic acid is a bioactive polyphenolic compound naturally occurring as secondary metabolite in many plant taxa that can inhibit the Keap1 to accumulate the Nrf2 in the nucleus and act on the ARE to produce target proteins, which in turn may alleviate the effects of ROS on neuronal cells in neurodegenerative diseases [126]. The Epigallocatechin gallate, a plant compound found mainly in green tea, acts as a natural catechin with antioxidant properties. It activates the Deap-1/Nrf2 pathway to inhibit oxidative stress and minimize high levels of reactive oxygen species, thus showing clinical applications in AD [127]. Curcumin, a phytopolylphenol pigment isolated from the plant Curcuma longa, user four different ways to stimulate the Nrf2 signaling pathway, including inhibition of Keap1, affecting the upstream mediators of Nrf2, influencing the expression of Nrf2 and target genes, and, finally, improving the nuclear translocation of Nrf2 [128]. Allicin, one of the main active compounds in garlic, has a dual neuroprotective mechanism through its ability to suppress oxidative stress by promoting Nrf2 levels and its potential to inhibit caspase-3 and prevent apoptosis [129], and the cognitive performance of patients suffering from neurological diseases can be improved by the administration of it [130]. β-Carotene has also been shown to be a potent inhibitor of oxidative stress and inflammation in vitro and in vivo. Treatment with β-carotene effectively suppressed the expression of proinflammatory cytokines and restored suppressed protein expressions of Nrf2 and HO-1 [131]. Another plant worth mentioning is Moringa (M. oleifera), a drought-resistant tree native to India, whose bioactive compounds enhance the brain’s antioxidant defenses, reduce inflammation, and improve neurotransmitter levels, showing potential therapeutic applications in neurodegenerative disorders, as its major isothiocyanate (MIC-1) can drastically reduce inflammatory cytokines in vitro and positively regulate Nrf2 [132].

N-Acetyl-Cysteine (NAC) residues function as key regulators in numerous cellular pathways and play critical roles at various structural and regulatory sites, including the modulation of Nrf2 activity. Notably, the oxidative modification of critical cysteine residues in Keap1 disrupts the Keap1-Cullin-3 complex within the ubiquitination pathway, preventing Nrf2 ubiquitination. This activates the antioxidant response, positioning Nrf2 as a promising therapeutic target for neurodegenerative diseases [133].

While the use of multiple compounds capable of activating Nrf2 in neurons seems promising in AD, it is important to note that Nrf2 also activates several oncogenes [134]. When there is persistent uncoupling between Keap1 and Nrf2, it can result in continuous activation of Nrf2, disrupting the normal regulation of cell proliferation. This uncontrolled activation may confer a survival advantage to cells under stressful conditions, but it also promotes carcinogenic characteristics [135]. Therefore, while compounds that activate Nrf2 have the potential to be beneficial in neurodegenerative diseases such as Alzheimer’s, it is also critical to consider the risks associated with prolonged Nrf2 activation, especially in the context of oncogenesis.

**Table 3 antioxidants-14-00121-t003:** Nrf2 activator compounds.

Compound	Mechanism of Action	Limitations	Efficacy
DJ-1	Acts as an oxidative stress sensor and regulates Nrf2 transcription, protecting against oxidative stress [115].	Pathogenic mutations associated with Parkinson’s disease [136].	Preclinical [136]
Compound B (DJ-1 inhibitor)	Potentiates Nrf2 activation through the PI3K/Akt pathway by binding to the sulfinated form of DJ-1 [116].	Requires further studies to validate efficacy and safety [137].	Preclinical [137]
ND-13	Peptide derived from DJ-1, activates the Nrf2 pathway and protects against neurotoxins in Parkinson’s and other diseases [118].	Limited studies in vitro and in vivo [117].	Preclinical [117]
Omaveloxolone (RTA408)	Activates Nrf2 and increases NQO1 expression, restoring damaged neurons and reducing Aβ aggregates in AD models [120].	Still under clinical investigation with limited specific applications [119].	Clinical (approved for Friedreich’s Ataxia) [119]
Dl-3-N-Butylphthalide	Inhibits NLRP3 by suppressing TXNIP-NLRP3 interaction, reducing inflammation [121].	Clinical efficacy not fully established in AD [122].	Preclinical [122]
Dimethyl Fumarate	Blocks Keap1-Nrf2 interaction, activating the antioxidant response [123].	Approved for multiple sclerosis, but its use in AD is still under evaluation [123].	Clinical (approved for MS) [138]
Sulforaphane	Increases Nrf2 expression, reducing oxidative stress-induced cytotoxicity [124].	Needs more clinical evidence for use in AD [124].	Preclinical [124]
Resveratrol	Inhibits Aβ protein aggregation and modulates intracellular effectors involved in neuronal survival/death [125].	Limited clinical studies in AD [125].	Preclinical [125]
Ellagic Acid	Inhibits Keap1, accumulating Nrf2 in the nucleus and activating target genes to alleviate ROS effects in neurodegeneration [126].	Requires more studies to validate efficacy in neurodegenerative diseases [126].	Preclinical [126]
Epigallocatechin Gallate (EGCG)	Activates Deap-1/Nrf2 pathway to reduce oxidative stress and minimize ROS [127].	Few clinical studies in AD [127].	Preclinical [127]
Curcumin	Stimulates the Nrf2 pathway through Keap1 inhibition, improving Nrf2 nuclear translocation [127].	Low bioavailability and need for improved formulations [127].	Preclinical [127]
Allicin	Promotes Nrf2, suppresses oxidative stress, inhibits caspase-3, and prevents apoptosis [129].	Limited clinical efficacy in neurodegenerative diseases [130].	Preclinical [130]
β-Carotene	Suppresses oxidative stress and inflammation, restoring Nrf2 and HO-1 protein expressions [131].	Limited efficacy and possible adverse effects at high doses [139].	Preclinical [139]
Moringa (M. oleifera)	Enhances antioxidant defenses in the brain, reduces inflammation, and regulates neurotransmitter levels, potentiating Nrf2 [132].	Needs more clinical trials to confirm effectiveness [132].	Preclinical [132]
N-Acetyl-Cysteine (NAC)	Modulates Nrf2 activity by modifying critical cysteines in Keap1, inhibiting its ubiquitination and activating the antioxidant response [133].	Risks associated with high doses, such as renal toxicity [133].	Clinical (in other conditions) [133]

## 5. Discussion

AD has been the subject of study for a long time, and it is thanks to this that many significant advances have emerged that help us to better understand the disease. It is now known that neuroinflammation induced by aberrant Aβ and Tau aggregates play a key role in its progression. The scientific community has made considerable efforts to elucidate the mechanisms that trigger uncontrolled inflammation in the brain. In 2002, Jürg Tschopp’s team discovered a new complex, the inflammasome, which was responsible for the inflammation suffered by patients with gout or type 2 diabetes [140]. In 2008, there was a significant breakthrough in the study of AD, when numerous investigations began to indicate that inflammasome activation played a crucial role in the progression of AD in animal models. This marked an important milestone in the understanding of how the inflammasome may be involved in the pathological processes of this neurodegenerative disease.

NLRP3 plays a crucial role in the pathogenesis of AD and can be activated by the presence of hyperphosphorylated Aβ1–42 and Tau aggregates [71], two key pathological features in AD. Activation of the NLRP3 inflammasome triggers a number of inflammatory responses that can be detrimental if exacerbated and chronic; one such response is pyroptosis, a type of burst cell death, which contributes significantly to neuronal damage and disease progression.

Furthermore, one of the essential components of the NLRP3 inflammasome, the ASC, has the ability to oligomerize with other ASC molecules to form structures known as ASC specks [75]. These structures not only act as inflammatory signal amplifying platforms, but also have the ability to accelerate Aβ aggregation, similar to the propagation properties of prions [78]. This prion-like behavior of ASC specks facilitates the expansion and aggravation of amyloid deposits in the brain, exacerbating the cellular damage and cognitive dysfunction observed in AD patients. Autophagy is a vital cellular process responsible for degrading and recycling damaged or unneeded components within the cell, but, in AD, this mechanism is compromised. Failed autophagy of Aβ and Tau aggregates leads to their accumulation, contributing to synaptic dysfunction and neuronal death. This accumulation causes oxidative stress, inflammation, and mitochondrial dysfunction, and could therefore be one of the causes of inflammasome activation.

Another component susceptible to degradation has been shown to be the inflammasome. Autophagy of inflammasomes may target key components such as the ASC, facilitating its degradation. This may negatively modulate inflammasome activity and pyroptosis by limiting the formation and persistence of active inflammasomes. ASC autophagy represents a critical regulatory mechanism in the inflammatory response, with potential implications in the pathogenesis of Alzheimer’s disease. ASC degradation by the lysosome causes ASC specks, Tau, and Aβ proteins to continue to accumulate, leading to a chronic immune response [76]. Several molecular mechanisms interconnect the autophagy pathway with inflammasome activation, so this interaction is likely to be crucial in the pathogenesis of AD [141].

The search for effective strategies to mitigate inflammation in AD has identified three particularly promising areas of focus: effective degradation of ASC clusters, specific inhibition of NLPR3 (Table 2), and specific Nrf2 activation (Table 3). Taken together, these lines of research not only have the potential to provide further insight into the underlying mechanisms of the disease, but also have the potential to open up new therapeutic avenues that could help mitigate the devastating impact of inflammation on the progression of Alzheimer’s disease.

### 5.1. Potential of Nrf2 in AD

Nrf2 inducers emerge as promising therapeutic tools in this context due to their ability to mitigate oxidative stress and down-regulate inflammasome-driven inflammation. When activated, Nrf2 modulates the expression of antioxidant and anti-inflammatory genes, directly inhibits NLRP3 inflammasome formation, and reduces proinflammatory activity mediated by ROS. This dual functionality positions Nrf2 as a key target for addressing both oxidative damage and chronic inflammation in AD. Compounds that activate Nrf2 (Table 3), such as sulforaphane and dimethyl fumarate, have shown promise in preclinical models, and some are even in clinical trials for other neurodegenerative diseases, providing evidence of their potential to treat AD (Table 3).

The main challenge in activating Nrf2 for AD lies in the need to improve the bioavailability and precision in the activation of the mechanism, as prolonged or dysregulated activation could have undesirable side effects, such as oncogenesis. Moreover, the efficacy of some compounds, like sulforaphane, still requires clinical validation in AD. Nevertheless, Nrf2 activation strategies hold tremendous translational potential, not only because they target two of the most fundamental pathological mechanisms in AD, but also because some compounds are already in clinical stages, offering the possibility of viable treatments for the disease in the near future.

### 5.2. Potencial of NLRP3 in AD

As mentioned above, NLRP3 is a key component in inflammasome formation and plays a central role in the neurogenic inflammation that exacerbates AD. NLRP3 activation leads to the release of proinflammatory cytokines and the amplification of the inflammatory response in the brain, contributing to the neurodegeneration observed in the disease. Inhibiting NLRP3, as shown in compounds such as Dl-3-N-Butylphthalide, could provide an effective way to reduce inflammation and, therefore, slow the progression of AD. However, the clinical efficacy of NLRP3 inhibitors is still under investigation, which limits their immediate potential in clinical practice.

Despite these limitations, specific inhibition of NLRP3 stands out as a valuable strategy. Since chronic inflammation is one of the most destructive aspects of AD, advances in NLRP3 inhibition could offer significant relief from the inflammatory and neurodegenerative symptoms in patients. While both pathways hold remarkable potential, the challenges lie in optimizing dosing, selecting appropriate compounds, and mitigating potential side effects. In this regard, future clinical studies will be crucial not only to determine the safety of these therapies, but also their efficacy in treating AD.

## 6. Conclusions and Future Perspectives

The uncontrolled activation of these inflammatory complexes, triggered by the accumulation of Aβ and hyperphosphorylated Tau proteins, promotes neuronal inflammation, pyroptosis, and cognitive dysfunction. The ASC plays a central role in this process, forming ASC specks that not only amplify inflammatory responses, but also accelerate Aβ aggregation, exacerbating the pathology.

Thus, therapeutic application may affect these mechanisms:Reduction in oxidative damage and neuroinflammation: Activating Nrf2 can lower ROS levels, inhibit inflammasome-mediated inflammation, and promote neuronal survival.Prevention of ASC speck formation: By limiting NLRP3 inflammasome activation, Nrf2 inducers can disrupt the positive feedback loop that worsens inflammation and encourages Aβ aggregation.Improvement in preclinical models: Compounds such as dimethyl fumarate, β-hydroxybutyrate, and omaveloxolone have shown protective effects in animal models of AD, reducing both amyloid burden and inflammation.

While promising, Nrf2 inducers must be carefully evaluated to avoid adverse effects, such as persistent activation of oncogenic pathways, and to optimize their safety and efficacy for treating AD. Ideally, a treatment that not only activates the antioxidant pathways of Nrf2, but that also inhibits inflammation mediated by NLRP3 could offer a holistic approach that simultaneously targets two of the most detrimental factors in disease progression: oxidative damage and chronic inflammation.

Due to recent research that has highlighted the importance of post-PTMs in the regulation of inflammasome activation, further studies are needed to evaluate inflammasome PTMs as potential therapeutic targets, as well as investigating the presence and activation of ASC specks and the role of the inflammasome in non-immune cells, such as neurons. These research avenues could open new ways to address neuronal inflammation and slow disease progression.

In conclusion, strategies that modulate Nrf2 and NLRP3 offer a promising future for AD therapy, but further research is required to explore these new possible approaches to become standard new treatments for the disease.

## Figures and Tables

**Figure 1 antioxidants-14-00121-f001:**
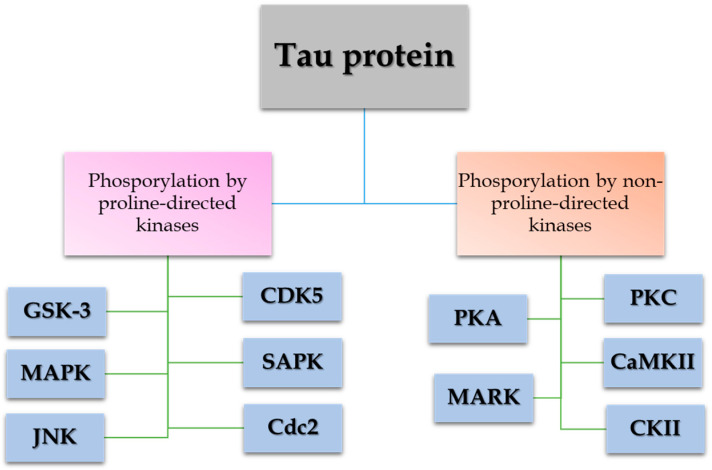
Phosphorylation of Tau at serine and threonine residues by proline-directed and non-proline-directed kinases. Proline-directed kinases, such as glycogen synthase kinase-3 beta (GSK-3β) and cyclin dependent kinase 5 (CDK5), are involved in the aberrant hyperphosphorylation of Tau and the progression of Alzheimer’s disease.

**Figure 2 antioxidants-14-00121-f002:**
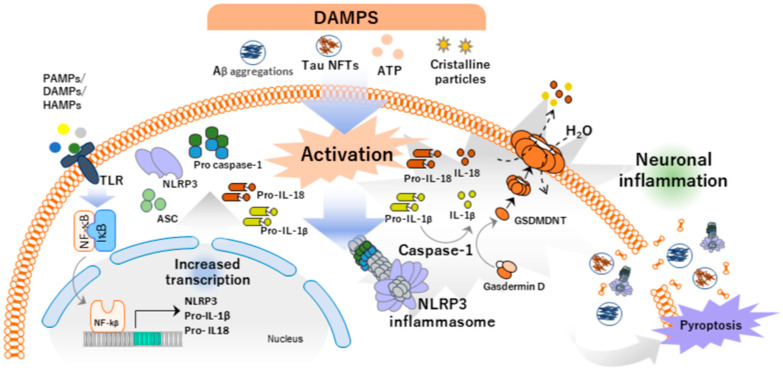
NLRP3 activation flowchart. The activation of the NOD-like receptor NLR family pyrin domain containing 3 (NLRP3) inflammasome occurs in two distinct steps. In the first step, Toll-like receptors (TLRs) are triggered by damage-associated molecular patterns (DAMPs), pathogen-associated molecular patterns (PAMPs), or hyaluronan-binding glycoproteins (HAMPs). This activation results in the translocation of nuclear factor kappa B (NF-κB) to the nucleus, where it enhances the transcription of inflammasome components, including pro-interleukin-1β (pro-IL-1β) and pro-interleukin-18 (pro-IL-18). In the second step, a secondary signal, typically derived from DAMPs such as adenosine triphosphate (ATP), Tau protein intracellular neurofibrillary tangles (NFTs), or amyloid-beta aggregates, induces the oligomerization of the NLRP3. The formation of these pores triggers inflammation by releasing mature interleukins and allows for water to enter the cell. Finally, pyroptosis occurs, and all cytoplasmic contents are released. As a consequence, this process promotes neuronal inflammation in Alzheimer’s disease (AD).

**Figure 3 antioxidants-14-00121-f003:**
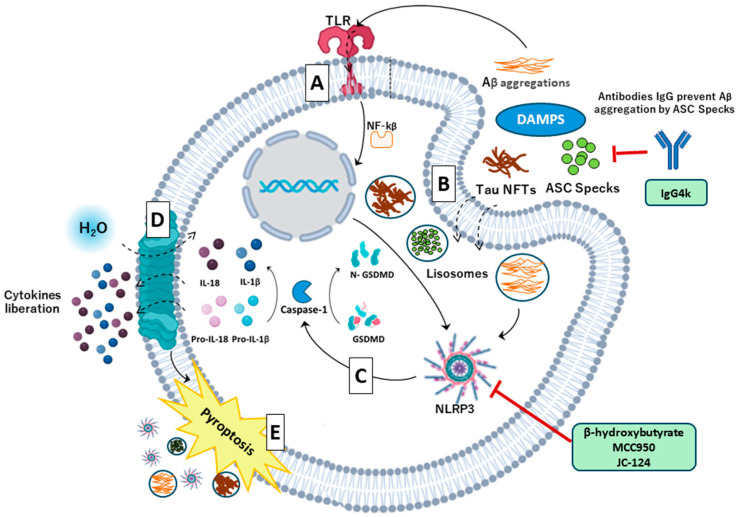
Molecular mechanisms that regulate death by pyroptosis induced by the accumulation of AB and Tau in Alzheimer’s disease (AD). (**A**) The β-amyloid (Aβ) aggregates (DAMPs) stimulate TLR receptors and the translocation of nuclear factor kappa B (NF-κB) to the cell nucleus, which, in turn, increases the transcription of the inflammasome components and expression of pro-interleukin-1β (pro-IL-1β) and pro-interleukin-18 (pro-IL-18). (**B**) Aberrant AD proteins are phagocytosed by microglia. The phagocytosed proteins are taken to the lysosome. There, they can disrupt the lysosomal membrane and cause assembly of the inflammasome. (**C**) Activation of the inflammasome leads to cleavage of pro-caspase 1 to caspase 1. Caspase 1 cleaves gasdermin D, which allows for the formation of the lysis pore. Caspase 1 also cleaves pro-IL-1B and pro-IL-18 to IL-1B and IL-18. (**D**) The insertion of the cleaved gasdermin N-terminal fragment into the plasma membrane creates oligomeric pores and allows for the release of proinflammatory products such as IL-1β and IL-18 to the extracellular space. Pore formation also induces water influx into the cell, cell swelling, and osmotic cell lysis, which induces further inflammation and hypertension by releasing more inflammatory products from the intracellular space. (**E**) As a consequence of all of the above processes, the osmotic pressure increases and normal cellular activities cease, and finally the cell undergoes an explosive death called pyroptosis.

**Table 2 antioxidants-14-00121-t002:** NLRP3 or ASC speck inhibitory compounds.

Compound	Mechanisms of Action	Evidence of Positive Effects on AE	Stage of Development
β-hydroxybutyrate	It inhibits caspase-1 cleavage, hinders the decrease in intracellular K^+^ efflux in macrophages, decreases microgliosis, and prevents oligomerization of ASCs and the appearance of ASC specks [91,92,93].	Yes, in 5XFAD mice [91]	Preclinical development [91].
IC 100 (IgG4κ)	IL-β decreases, ASC uptake by IC 100 prevents the inflammasome from polymerizing with ASC, and ASC specks are formed [81,94].	No (used as marker in EA).	Not investigated.
MCC950	It blocks canonical and non-canonical NLRP3 activation, inhibits procaspase-1 cleavage, ASC specks form, and reduces Tau hyperphosphorylation [95,96,97].	Yes, in APP/PS1 mice [96] and PS19 transgenic mice with the Tau P301S mutation on a C57BL/6J background [97].	Preclinical Development [96,97].
JC-124	Blocks NLRP3 formation and caspase-1 activation, inhibits β-secretase [98].	Yes, in TgCRND8 mice [98].	Preclinical development [98].
NLRP3 protein kinase A	Phosphorylates at Ser295 attenuating the ATPase activity of NLRP3 and causing its inhibition, leads NLPR3 to degrade via ubiquitination [87,88].	No.	Not investigated.

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
