# Peer review of "Inflammasomes in Alzheimer’s Progression: Nrf2 as a Preventive Target"

_antioxidants, 2025, doi:10.3390/antiox14020121_

Round 1
Reviewer 1 Report
The manuscript "Inflammasomes in Alzheimer’s Progression: Nrf2 as a Preventive Target" offers a comprehensive overview of Alzheimer’s Disease (AD), particularly emphasizing inflammasome activation and the therapeutic potential of Nrf2. The paper is dense, with scientific details and relevant findings. However, there are notable areas for refinement to enhance clarity, depth, and coherence.
1. Abstract (Lines 15-29). The abstract is overly complex, with jargon and long sentences. Revise "the autophagic mechanism of inflammasomes seems to be defective". Clarify the therapeutic potential of Nrf2 early on to frame its importance.
2. (Lines 48-56) The historical section on Alzheimer’s lacks transition and connection to the disease's molecular basis.
3. (Lines 34-47) Risk factors are listed without explanation of their mechanistic link to AD. Briefly explain how hypertension, diabetes, and obesity contribute to AD pathophysiology.
4. (Lines 57-113) Repetition of Aβ formation pathways in different sections makes the narrative circular.
5. (Lines 127-140) The molecular mechanism of Tau hyperphosphorylation lacks clarity on kinase interactions. Provide a diagram summarizing the key steps of Tau phosphorylation and its outcomes.
6. (Lines 197-299) The inflammasome section introduces NLRP3 without adequately tying it to AD progression. Add a paragraph explicitly connecting inflammasome activation to neuronal pyroptosis and cognitive decline.
7. No figures or tables visually represent complex mechanisms (e.g., inflammasome activation or Nrf2 pathways). To aid comprehension, include a flowchart for NLRP3 activation and its downstream effects.
8. (Lines 305-335) Studies on pyroptosis and inflammasomes are listed without comparative analysis. Compare the methodologies and findings of the studies to highlight the consensus and gaps in the literature.
9. (Lines 358-410) The role of ASC specks is described in isolation without discussing their potential therapeutic implications. Conclude the section with a summary of therapeutic opportunities targeting ASC specks.
10. (Lines 412-445) Expand on how post-translational modifications (PTMs) influence the crosstalk between inflammasomes and other cellular pathways.
11. (Lines 447-455) The therapeutic strategies table lacks adequate references and context. Include references for each compound and briefly describe its stage of development (preclinical/clinical).
12. (Lines 575-645) The Nrf2 Activators section lists activators without prioritizing or evaluating their effectiveness. Rank the compounds based on preclinical or clinical efficacy and address potential limitations.
13. (Lines 655-701) The discussion reiterates results without providing new insights or broader implications. Incorporate a subsection on the translational potential of inflammasome and Nrf2-targeted therapies.
14. (Lines 709-727) Future perspectives are too brief and lack specificity. Propose specific research directions or experimental designs that address gaps in current knowledge.
Author Response
The manuscript "Inflammasomes in Alzheimer’s Progression: Nrf2 as a Preventive Target" offers a comprehensive overview of Alzheimer’s Disease (AD), particularly emphasizing inflammasome activation and the therapeutic potential of Nrf2. The paper is dense, with scientific details and relevant findings. However, there are notable areas for refinement to enhance clarity, depth, and coherence.
- Abstract (Lines 15-29). The abstract is overly complex, with jargon and long sentences. Revise "the autophagic mechanism of inflammasomes seems to be defective". Clarify the therapeutic potential of Nrf2 early on to frame its importance.
Response: The requested modifications have been implemented in the text.
- (Lines 48-56) The historical section on Alzheimer’s lacks transition and connection to the disease's molecular basis.
Response: The molecular basic of the diseases is descrived follewing, this paragrhts just explain the origin of the diseases discovering.
- (Lines 34-47) Risk factors are listed without explanation of their mechanistic link to AD. Briefly explain how hypertension, diabetes, and obesity contribute to AD pathophysiology.
Response: The requested modifications have been implemented in the text.
- (Lines 57-113) Repetition of Aβ formation pathways in different sections makes the narrative circular.
Response: The requested modifications have been implemented in the text.
- (Lines 127-140) The molecular mechanism of Tau hyperphosphorylation lacks clarity on kinase interactions. Provide a diagram summarizing the key steps of Tau phosphorylation and its outcomes.
Response: The diagram have been implemented in the manuscript.
- (Lines 197-299) The inflammasome section introduces NLRP3 without adequately tying it to AD progression. Add a paragraph explicitly connecting inflammasome activation to neuronal pyroptosis and cognitive decline.
Response: The text has been expanded as required to include the requested information.
- No figures or tables visually represent complex mechanisms (e.g., inflammasome activation or Nrf2 pathways). To aid comprehension, include a flowchart for NLRP3 activation and its downstream effects.
Response: The new figure has been implemented in the manuscript.
- (Lines 305-335) Studies on pyroptosis and inflammasomes are listed without comparative analysis. Compare the methodologies and findings of the studies to highlight the consensus and gaps in the literature.
Response: The requested modifications have been implemented in the text.
- (Lines 358-410) The role of ASC specks is described in isolation without discussing their potential therapeutic implications. Conclude the section with a summary of therapeutic opportunities targeting ASC specks.
Response: Additional content has been added to the manuscript in response to the reviewers' suggestions
- (Lines 412-445) Expand on how post-translational modifications (PTMs) influence the crosstalk between inflammasomes and other cellular pathways.
Response: Relevant text has been included to address the specific concerns raised by the reviewers.
- (Lines 447-455) The therapeutic strategies table lacks adequate references and context. Include references for each compound and briefly describe its stage of development (preclinical/clinical).
Response: The requested information has been implemented in the table.
- (Lines 575-645) The Nrf2 Activators section lists activators without prioritizing or evaluating their effectiveness. Rank the compounds based on preclinical or clinical efficacy and address potential limitations.
Response: The requested changes have been implemented and a new table with the lists of Nrf2 activators has been included in the manuscript with the requested information.
- (Lines 655-701) The discussion reiterates results without providing new insights or broader implications. Incorporate a subsection on the translational potential of inflammasome and Nrf2-targeted therapies.
Response: The requested modifications have been implemented in the text.
- (Lines 709-727) Future perspectives are too brief and lack specificity. Propose specific research directions or experimental designs that address gaps in current knowledge.
Response: The requested modifications have been implemented in the text.

Reviewer 2 Report
This review is very relevant contribution to the field as it describes in detail the neuroinflammation and especially the role of ASC specks in progression of Alzheimer's disease. A recent study has demonstrated the utility of ASC specks as a fluid biomarker of Alzheimer's (https://doi.org/10.1038/s41467-024-53547-0), thus, the importance of the role of ASC specks in progression Alzheimer's disease is confirmed (I recommend to integrate this reference into review text). The link between ASC specks and Nrf2 described in this review offer a pharmacological strategies, as increasing number of Nrf2 drugs are becoming available, with some of them already being FDA approved. To my knowledge, the link between ASC specs and Nrf2 signaling in respect to Alzheimer is relatively new field, thus, this review is is timely and relevant.
There are some spelling errors and text does not seem smooth at some places, but in general, the review is well written.
I believe, that another more detailed Figure that summarizes the role of the ASC specks and Nrf2 in AD will benefit the manuscript. The is one figure under the abstract, but a more detailed figure focusing on discussion and conclusions would benefit in explaining rather complex material to the reader. But this is just a suggestion.
Minor remarks:
Line 20- abbreviation "NLR" should be spelled out and defined.
Line 72- please specify N7 receptor more specifically- as nicotinic acetylcholine receptor.
Line 122- remove the repetition of "cytosolic Ca".
lines 170-173- remove the repetitions.
Line 174- i believe Manning FC is a missing reverence for the statement.
Table 2, Ion charge is missing in "intracellular K efflux".
Lines 513 and 515, same as previous.
Author Response
This review is very relevant contribution to the field as it describes in detail the neuroinflammation and especially the role of ASC specks in progression of Alzheimer's disease. A recent study has demonstrated the utility of ASC specks as a fluid biomarker of Alzheimer's (https://doi.org/10.1038/s41467-024-53547-0), thus, the importance of the role of ASC specks in progression Alzheimer's disease is confirmed (I recommend to integrate this reference into review text). The link between ASC specks and Nrf2 described in this review offer a pharmacological strategies, as increasing number of Nrf2 drugs are becoming available, with some of them already being FDA approved. To my knowledge, the link between ASC specs and Nrf2 signaling in respect to Alzheimer is relatively new field, thus, this review is is timely and relevant.
There are some spelling errors and text does not seem smooth at some places, but in general, the review is well written.
Response: Todo el manuscrito ha sido revisado y editado a fondo para garantizar la corrección gramatical y una legibilidad óptima.
I believe, that another more detailed Figure that summarizes the role of the ASC specks and Nrf2 in AD will benefit the manuscript. The is one figure under the abstract, but a more detailed figure focusing on discussion and conclusions would benefit in explaining rather complex material to the reader. But this is just a suggestion.
Response: The requested modifications have been implemented in the manuscript and a new figure has been added.
Detail comments
Minor remarks: Belen y MM.
Line 20- abbreviation "NLR" should be spelled out and defined.
Line 72- please specify N7 receptor more specifically- as nicotinic acetylcholine receptor.
Line 122- remove the repetition of "cytosolic Ca".
lines 170-173- remove the repetitions.
Line 174- i believe Manning FC is a missing reverence for the statement.
Table 2, Ion charge is missing in "intracellular K efflux".
Lines 513 and 515, same as previous.
Response: All modifications suggested by the reviewers have been incorporated into the text.

Round 2
Reviewer 1 Report
All the requested modifications have been implemented in the text.
-